# Recent Advances in the Analysis of Vitamin D and Its Metabolites in Food Matrices

**Bárbara Socas-Rodríguez [1],\*, Margareta Sandahl [1], Cecilia Holm [2] and Charlotta Turner [1]**

[1] Department of Chemistry, Centre for Analysis and Synthesis, Lund University, P.O. Box 124, 221-00 Lund, Sweden; Margareta.Sandahl@chem.lu.se (M.S.); Charlotta.Turner@chem.lu.se (C.T.)
[2] Department of Experimental Medical Science, Faculty of Medicine, Lund University, P.O. Box 124, 221-00 Lund, Sweden; cecilia.holm@med.lu.se
\* Correspondence: barbara.socas_rodriguez@chem.lu.se

**Abstract:** Vitamin D and its analogues are fat-soluble vitamins that carry out important functions in human and animal organisms. Many studies have pointed out the relationship between the deficiency of these substances and the development of both skeletal- and extra-skeletal diseases. Although vitamin D is fundamentally derived from the bio-transformation of its precursor, 7-dehydrocholesterol, through the action of UV-B radiation in the skin, dietary intake also plays an important role in the regulation of its status in an organism. For this reason, the application of reliable methodologies that enable monitoring the content of vitamin D and its analogues in food and supplements constitutes an aspect of special relevance to establish adequate habits, which avoid the deficiency of these substances in organisms and, consequently, the appearance of related diseases. The use of chromatographic techniques in combination with conventional and novel sample pre-treatments has become a suitable strategy to achieve this aim. This review compiles the most relevant methodologies reported in the last ten years for vitamin D analogues analysis in food matrices. Particular attention has been paid to provide a general overview of the most suitable approaches in terms of reliability, sensitivity and simplicity, used in the field of food analysis.

**Keywords:** vitamin D; vitamin D metabolites; food analysis; sample preparation; chromatographic techniques; saponification; conventional extraction; miniaturized techniques

## 1. Introduction

Vitamin $D_3$ (cholecalciferol) ($D_3$) and vitamin $D_2$ (ergocalciferol) ($D_2$) are fat-soluble vitamins with a steroidal structure. They play an important role in calcium and phosphate absorption, as well as bone metabolism, and have many other pleiotropic functions in humans and other species [1]. $D_3$ is fundamentally derived from the transformation of its precursor, 7-dehydrocholesterol (7-DHC), present in the skin, through the action of ultraviolet radiation B (UV-B, 290–300 nm). It is converted into previtamin $D_3$ (Pre-$D_3$) which finally is thermally isomerized to $D_3$. However, synthesis in the skin is not the only source of $D_3$. In fact, around 10–20% of this substance is obtained through dietary intake by absorption in the intestine, although it also depends on the area and the season around the world [2]. In contrast, $D_2$ is principally obtained from the consumption of edible plants, fungi or yeast, which also transform previtamin $D_2$ (Pre-$D_2$) into $D_2$ by UV radiation action [1]. Both $D_3$ and $D_2$ are inactive substances that are transported through the blood stream to the liver, where the most stable metabolites, 25-hydroxyvitamin D (25-OHD) forms, are generated. Those are further hydroxylated into the most active metabolite, 1,25-dihydroxyvitamin D (1,25-OHD), in kidney and other organs such as breast, colon, brain, pancreas and prostate. Additionally, other less active forms, as, for example, 24,25-dihydroxyvitamin D (24,25-$(OH)_2$D) or 1-hydroxyvitamin D (1-OHD) and their

epimeric forms, are also generated [3]. Apart from hydroxylated metabolites, it is believed that ester forms are produced in different tissues of the organism, although the routes and tissues in which these metabolites are obtained and the roles that they play in the organism are not very well-known yet. Despite the fact that $D_2$ is not produced by mammals, the active forms of $D_2$ and $D_3$ are considered equivalent, although this is an issue of controversy [4].

Currently, there is a global concern about the status of vitamin D and its metabolites in the population since it has been reported that insufficiency and deficiency in these vitamins in humans is linked not only to the development of bone diseases, such as rickets and osteoporosis, but also to many extra-skeletal diseases, including diabetes, hypertension, cardiovascular diseases, multiple sclerosis, psoriasis, Crohn's disease, neuropsychiatric illnesses or even breast and colon cancer [3,5–7]. This is why increasing the consumption of foods which contain these vitamins, with a daily intake of 5–20 μg in adults, is essential to tackle this problem, especially for the population that lives at high latitudes, where the subcutaneous synthesis of vitamin D is absent during winter months. However, it should be emphasized that the recommendations for daily intake vary considerably between different population groups and advisory guidelines [3]. Besides, intake of excessive doses can cause other health problems, such as hypercalcemia, hypercalciuria or hyperphosphatemia [8].

Fatty fish, fish oil, meat or egg yolk are generally considered to be food items with a high content of vitamin D, although it can also be found in fruit, edible plants and seeds [8–10]. Nevertheless, the content of these substances in the diet is limited and their stability in food matrices is also questioned under certain conditions such as light, heat or oxidant conditions [1]. For this reason, there are international recommendations for food fortification including $D_3$, $D_2$ and 25-OHD analogues [11]. In this sense, diverse studies have been carried out in order to enrich widely consumed foodstuffs by the addition of these fat-soluble vitamins or by "bio-addition" enhancing the bio-production process in the products [2,11]. In the same direction, the US Food and Drug Administration (FDA) revised the food labelling guidelines in 2016 to make vitamin D content required information in conventional food and supplement packages with the aim of promoting vitamin D awareness among consumers [12]. Additionally, the Codex Alimentarius recommends that food for infants and young children is fortified with $D_2$ and $D_3$ [13], and many countries, including United Kingdom, Canada, Australia, United States, Finland, Denmark and Ireland, have mandatory or voluntary food fortification regulations [14]. This is why a strict evaluation of the presence of vitamin D and its metabolites in food items is an issue of special relevance for the scientific community and also for general society, in order to get an accurate knowledge of the population intake.

In this regard, the determination of vitamin D and its metabolites in food matrices has been traditionally carried out using bio-assay methods. However, these kinds of techniques are time-consuming and, additionally, cross-reactions occur as a result of the similar structures of the different analogues, hampering the analysis of the different individual bioactive compounds [10]. This is why the application of chromatographic techniques has become one of the most adequate alternatives since the 1980s because such applications allow the selective separation and analysis of each compound individually. Indeed, the official methods for determining $D_3$ and $D_2$ in dried skimmed milk and in general commodities, proposed by the International Dairy Federation (IDF) [15] and the European Commission (EC) [16], respectively, are based on the use of liquid chromatography (LC) combined with UV detection. In addition, the Association of Official Analytical Chemists (AOAC) has also proposed other methods for the determination of these two analytes in nutritional formulas using LC hyphenated with tandem mass spectrometry (MS/MS) [17,18], after the proposal of diverse methodologies based on high-performance liquid chromatography (HPLC)-UV analysis [19–21]. Apart from that, the Codex Alimentarius proposed in 2018 the endorsement of the AOAC Official Method 2016.05 as type II—the most suitable method—for the analysis and sampling of vitamin D in infant formulas displacing the EN 12821, based on HPLC-UV analysis, as type III, only necessary for verification in case of controversy. However, it is worth mentioning that most of the applications developed so far in this field have been focused on the determination of $D_2$ and $D_3$ forms [8,18,22–39] and only a few applications have

been performed for the analysis of hydroxylated and precursor forms [4,7,9,10,40–42], whereas the determination of ester forms has not been reported up to date in these kinds of matrices (see Table 1).

This review article is aimed at providing an overview of the most recent advances in the analysis of vitamin D and its analogues in food matrices, paying special attention to the most promising extraction methods and their combination with chromatographic techniques for the suitable determination of this group of fat-soluble vitamins.

**Table 1.** Structure of vitamin analogues commonly analyzed in food matrices.

| Name | Structure | Molecular Formula | Molecular Weight (g/mol) | Name | Structure | Molecular Formula | Molecular Weight (g/mol) |
|---|---|---|---|---|---|---|---|
| D$_3$ |  | C$_{27}$H$_{44}$O | 384.65 | 1OH-D$_2$ |  | C$_{28}$H$_{44}$O$_2$ | 412.65 |
| D$_2$ |  | C$_{28}$H$_{44}$O | 396.66 | 24,25-(OH)$_2$D$_3$ |  | C$_{27}$H$_{44}$O$_3$ | 416.64 |
| 25-OHD$_2$ |  | C$_{28}$H$_{44}$O$_2$ | 412.65 | 1,25-(OH)$_2$D$_2$ |  | C$_{28}$H$_{44}$O$_3$ | 428.70 |
| 25-OHD$_3$ |  | C$_{27}$H$_{44}$O$_2$ | 400.64 | 1,25-(OH)$_2$D$_3$ |  | C$_{27}$H$_{44}$O$_3$ | 416.64 |
| 1-OHD$_3$ |  | C$_{27}$H$_{44}$O$_2$ | 400.64 | | | | |

## 2. Recent Applications of Sample Preparation

The complexity of food matrices containing polysaccharides, proteins or lipids, among many other components, as well as the low concentration levels at which vitamin D analogues are present in such samples, make necessary the application of effective sample preparation procedures before the separation and determination of these substances using chromatographic techniques. However, their chemical instability, dealing with certain factors such as light, oxygen or pH, and their easy isomerization at high temperature make this step and the conditions at which it is developed, decisive aspects in the determination process [3].

In general terms, conventional solid-liquid extraction (SLE) and liquid-liquid extraction (LLE) have been the most widely applied techniques for the extraction of this vitamin and its metabolites from food matrices. However, solid-phase extraction (SPE) and preparative chromatography combined with SLE and LLE have also been applied as a common sample pre-treatment in this field. Additionally, a few applications of miniaturized techniques have been performed with the same aim, although to a lesser extent. In most cases, saponification in the presence of antioxidant agents was carried out at the beginning of the process in order to favor the separation of the analytes from the matrix and remove interferences [3], although other hydrolysis procedures or deproteinization have also been applied to accomplish this aim.

The application of saponification is focused on the chemical degradation of complex lipids present in this type of sample that can preclude the determination of vitamins. This procedure is carried out

by a hydration reaction, in which ester bonds are broken by the action of an alkali agent, obtaining hydrosoluble free fatty acids and glycerol that can be easily removed in the aqueous phase [43,44]. With this aim, a certain concentration of KOH in ethanol (EtOH) was used in most cases as a strong alkali agent to carry out the reaction, and a high temperature (around 60–80 °C) was applied in order to reduce the process time [7,24–26,30,33,34,41] and avoid overnight reactions, which were necessary when room temperature was applied [9,10,42]. However, some authors have reported short reaction times (30–60 min) under room temperature with good results for the evaluation of milk-based formulas, fruit juice and vegetable beverages, after a thorough optimization of time and temperature factors [28,29]. In this sense, the work carried out by Kwak et al. [29] should be highlighted. In this case, saponification using 150 mg of KOH, 150 mg of $NH_3$ and 2.4 g of NaCl was performed during 5 min at room temperature, after dilution and subsequently extraction using 10 mL of isopropanol (IPA) for the extraction of $D_3$ from milk-based formulas. The extract was then evaporated and injected in a HPLC-MS/MS system after reconstitution. The methodology was compared with the AOAC official method [17], in which high-temperature (75 °C) saponification during 30 min was necessary for the determination of $D_2$ and $D_3$ in infant formula and adult nutritionals.

Additionally, due to the instability of these vitamins during the procedure [3], antioxidant agents are usually added to the reaction media to assure their correct determination. In this sense, ascorbic acid [9,24,34,35], pyrogallol [10,26], sodium ascorbate [7,36,42] and butylated hydroxytoluene (BHT) [23,28] have been the most commonly antioxidant agents applied with this aim.

## 2.1. Conventional Extraction Techniques

As previously indicated, despite the fact that conventional SLE and LLE are complex and time consuming, they have been the most widely applied extraction techniques for the evaluation of food matrices (see Table 2). Nevertheless, other alternative procedures have also been sporadically applied. In this sense, the use of the dilute-and-shoot strategy should be remarked. As an example, Byrdwell et al. [27] applied this pre-treatment in combination with HPLC-mass spectrometry (MS) and HPLC-UV determination for the evaluation of $D_3$ in dietary supplements. The procedure avoids the application of complex steps, limiting the method to sample weighing, addition of an internal standard (IS) and dilution with the adequate solvent [43]. In this case, oil contained in the supplement gel caps was diluted to 100 mL using methanol (MeOH) and dichloromethane (DCM) in the composition MeOH/DCM (6/4, *v/v*), and was directly injected in the chromatographic system. This study demonstrated the suitability of the methodology for the evaluation of not only $D_3$, but also triacylglycerols in supplement gel caps and oil samples in combination with both LC-MS and LC-UV systems, showing that UV detection provided lower sample-to-sample relative standard deviation than atmospheric pressure chemical ionization (APCI)-MS detection.

**Table 2.** Recent applications of conventional extraction techniques for the analysis of vitamin D-related compounds in food matrices.

| Analytes | Matrix (Amount) | Sample Pre-Treatment | Determination Technique | Recovery % | LOD | Comments | Reference |
|---|---|---|---|---|---|---|---|
| $D_2$, $D_3$, 25-OHD$_2$, 3-epi-25-OHD$_2$, 25-OHD$_3$, 3-epi-25-OHD$_3$ | Commercial breast milk (1 mL) | Deproteinization (40 °C, 1 h, EtOH (1 mL)); LLE 2 X (hexane/EtOAc (9/1, *v/v*) (2.5 mL)) | UHPSFC-(QqQ)-MS/MS | 70–107 | 0.02 µg/L [a] | -APCI (+) was used as ionization source. -NH$_4$HCOO in MeOH was used as make-up solvent. -Different columns were evaluated. -Derivatization was carried out using PTAD. -Deuterated compounds were used as ISs. -Comparison between saponification and deproteinization. -SRM was used for method validation. -SFC conditions: fluoro-phenyl column, mobile phase (MeOH/NH$_4$HCOO/H$_2$O/CO$_2$), 45 °C. | [4] |
| $D_3$ | Rice (1 g) | SLE (hexane (4 mL), 5 min under N$_2$) | HPLC-UV | - | - | -Vitamin A, E and amino acids and pesticides were also evaluated. -LC conditions: C$_{18}$ column, mobile phase (ACN/H$_2$O), 30 °C. -Occurrence: 13.8–28.6 mg/kg. | [22] |
| $D_2$, $D_3$ | Infant formula, cereals, adult nutritionals, mixed meals (2 g) | Hydrolysis (α-amylase (50 mg): 45 °C, 30 min, papain solution (5 mL)); LLE (acidified MeOH (20 mL), BHT in isooctane (10 mL)) | UHPSFC-(QqQ)-MS/MS | 90–110 | 40 pg [a] | -APCI (+) was used as ionization source. -NH$_4$HCOO in MeOH was used as make-up solvent. -SRM was used for method validation. -Vitamin K, A and D were also evaluated. -Derivatization was carried out using PTAD. -Isotopically labelled compounds were used as surrogate ISs. -SFC conditions: 1-AA column, mobile phase (MeOH/NH$_4$HCOO/H$_2$O/CO$_2$), 45 °C. | [23] |

**Table 2.** *Cont.*

| Analytes | Matrix (Amount) | Sample Pre-Treatment | Determination Technique | Recovery % | LOD | Comments | Reference |
|---|---|---|---|---|---|---|---|
| $D_2$, $D_3$ | Fortified bread, bovine milk, infant formula (4 g) | **Milk and infant formulas:** VA-LLE (MeOH/$H_2O$/isooctane (6.25/1.25/2.50, *v/v/v*), 2 min). **Bread:** homogeneization (EDTA (2 mL), ascorbic acid (100 mg), vortex), saponification (KOH in water, 60 °C, 30 min), LLE (MeOH/isooctane, (2/1, *v/v*), (30 mL)) | HPLC-(QqQ)-MS/MS | - | - | -ESI (+) was used as ionization source. -Deuterated $D_3$ was used as surrogate IS. -Comparison of $D_2$ and $D_3$ supplement efficiency. -LC conditions: $C_{18}$ column, mobile phase (MeOH/$H_2O$/$NH_4HCOO$), 40 °C. | [24] |
| $D_3$ | Finfish, shellfish (30 g) | Folch method ($CHCl_3$/MeOH, (2/1, *v/v*) (450 mL)); saponification (KOH in MeOH, 30 min, refluxing): LLE (petroleum ether, (50 mL)) | HPLC-UV | - | - | -Mineral and other fat-soluble vitamins were also evaluated. -LC conditions: $C_{18}$ column, mobile phase (MeOH/ACN), -. -Occurrence: 7.72–23.28 µg/kg. | [25] |
| $D_2$, $D_3$ | Milk powder, infant formulas, nutritional formulas (1.8–21.0 g or 10 mL) | Saponification (KOH, pyrogallol, EtOH, 1 h, 70 °C); LLE (isooctane (10 mL)); washing ($H_2O$) | UHPLC-(QqQ)-MS/MS | 96–101 | 1.2–1.6 µg/kg | -ESI (+) was used as ionization source. -Derivatization was carried out using PTAD. -AOAC Official Method 2016.05. -Deuterated standards were used as ISs. -LC conditions: $C_{18}$ column, mobile phase (MeOH/$H_2O$/FA), 40 °C. | [26] |
| $D_3$, $D_2$, 25-OHD$_2$, 25-OHD$_2$, 24,25-(OH)$_2$D$_2$, 24,25-(OH)$_2$D$_3$, 1,25-(OH)$_2$D$_2$, 1,25-(OH)$_2$D$_3$ | Human, cow, mare, goat and sheep milk (4 mL) | Deproteinization (ACN (8 mL), vortex 2 min, room temperature (15 min)); LLE 2 X (hexane/DCM (1/4, *v/v*) (12 mL)) | HPLC-(QqQ)-MS/MS | 88–99 | 0.27–0.47 pM | -ESI (+) was used as ionization source. -Different columns were tested. -Derivatization was carried out using PTAD. -Comparison between saponification and deproteinization. -Deuterated standards were used as ISs -LC conditions: $C_{18}$ column, mobile phase (MeOH/$H_2O$/FA), -. | [39] |

**Table 2.** *Cont.*

| Analytes | Matrix (Amount) | Sample Pre-Treatment | Determination Technique | Recovery % | LOD | Comments | Reference |
|---|---|---|---|---|---|---|---|
| $D_3$ | Dietary supplements (-) | Dilute-and-shoot strategic (-) | HPLC-(TSQ)-MS, HPLC-(TSQ)-MS/MS, HPLC-UV | - | - | -APCI (+) was used as ionization source. -SIM and full scan modes were compared. -APCI and ESI were compared. -$D_2$ was used as IS. -Triacylglycerols were also evaluated. -LC conditions: hydrophilic endcapping-$C_{18}$ column, mobile phase (MeOH/ACN/DCM), 10 °C. | [27] |
| $D_2$, $D_4$, pre-$D_2$, pre-$D_4$, tachysterol$_2$, tachysterol$_4$, lumisterol$_2$, lumisterol$_4$, ergosterol, 22,23-dihydroergosterol | Oyster mushroom (5 g) | **Method 1:** Saponification/hydrolysis (KOH, sodium ascorbate, NaOH, 1 h, 80 °C, refluxing); LLE (diethyl ether (50 mL); EtOH/pentane (1/5, *v/v*) (60 mL), pentane (50 mL and 20 mL)). **Method 2:** Saponification/hydrolysis (KOH, sodium ascorbate, NaOH, 20 h, 18 °C, refluxing); LLE (diethyl ether (50 mL); EtOH/pentane (1/5, *v/v*) (60 mL), pentane (50 mL and 20 mL)). **Method 3:** US-LLE X 3 (pentane (100, 100 and 50 mL), 10 min, 13 °C) | HPLC-UV, HPLC-(QqQ)-MS/MS | 97 | 0.02–0.06 mg/kg | -APCI (+) was used as ionization source. -Analysis was carried out after UV-B treatment. -$D_3$ was used as surrogate IS. -Different extraction methods were compared. -LC conditions: hydrophilic endcapping-$C_{18}$ column, mobile phase (MeOH/$H_2O$/FA), 40 °C. | [7] |
| Pre-$D_3$, $D_3$ | Fortified cereals, cereals-based food, milk, fruit juice, yogurt, sun flower oil, egg yolk (5–30 g) | Saponification (KOH, $H_2O$, EtOH, 20 min, 80 °C); LLE (ciclohexane, (41 mL)) | 2D-HPLC-UV-(QqQ)-MS/MS | 96–105 | 0.5–0.8 µg/kg | -ESI (+) was used as ionization source. -Deuterated $D_3$ was used as IS. -SRMs were used for validation. -Pre-$D_3$, $D_3$ were analyzed together. -LC conditions: $C_8$ and $C_{18}$ columns, mobile phase (MeOH/ACN/$H_2O$/FA), 35 °C. | [41] |

**Table 2.** *Cont.*

| Analytes | Matrix (Amount) | Sample Pre-Treatment | Determination Technique | Recovery % | LOD | Comments | Reference |
|---|---|---|---|---|---|---|---|
| $D_2$, $D_3$ | Milk, infant formula, fruit juice, vegetable beverage (20–25 g) | Saponification (KOH, EtOH, BHT, 1 h, room temperature), LLE (2× hexane (50 mL)) | HPLC-DAD | - | 0.82–1.57 ng [b] | -Vitamin E was also evaluated. -$C_8$ and $C_{18}$ columns were compared. -LC conditions: $C_{18}$ column, mobile phase (MeOH/ACN), -. | [28] |
| $D_3$ | Milk-based formula (0.5 g) | Dilution ($H_2O$, 10 mL);VA-LLE (IPA (10 mL) with $(NH_4)_2SO_4$); saponification (KOH, NaCl, $NH_3$, 30 min, room temperature) | HPLC-(QqQ)-MS/MS | 93–110 | 0.84 µg/kg | -ESI (+) was used as ionization source. -SRMs were used for validation. -Comparison with official method. -LC conditions: $C_{18}$ column, mobile phase (MeOH/$H_2O$/$NH_4HCOO$), 40 °C. | [29] |

[a] limit of quantification (LOQ); [b] instrumental limit of detection (LOD); 1-AA:1-aminoanthracene; 2D-HPLC: two dimensional-high performed liquid chromatography; 3-epi-25-OHD$_2$: 3-epi-25-hydroxyvitamin $D_2$; 3-epi-25-OHD$_3$: 3-epi-25-hydroxyvitamin $D_3$; 25-OHD$_2$: 25-hydroxyvitamin $D_2$; 25-OHD$_3$: 25-hydroxyvitamin $D_3$; ACN: acetonitrile; AOAC: Association of Official Analytical Chemists; APCI: atmospheric pressure chemical ionization; BHT: butylated hydroxytoluene; $C_8$: octasylane; $C_{18}$: octadecylsylane; $D_2$: vitamin $D_2$; $D_3$: vitamin $D_3$; $D_4$: vitamin $D_4$; DAD: diode array detector; DCM: dichloromethane; EDTA: ethylenediaminetetraacetic acid; ESI: electrospray ionization; EtOAc: ethyl acetate; EtOH: ethanol; FA: formic acid; HPLC: high-performance liquid chromatography; IPA: isopropanol; IS: internal standard; LC: liquid chromatography; LLE: liquid–liquid extraction; LOD: limit of detection; MeOH: methanol; MS: mass spectrometry; MS/MS: tandem mass spectrometry; Pre-D$_2$: previtamin-$D_2$; Pre-D$_3$: previtamin-$D_3$; Pre-D$_4$: previtamin-$D_4$; PTAD: 4-phenyl-1,2,4-triazoline-3,5-dione; QqQ: triple quadrupole; SIM: selected ion monitoring; SLE: solid–liquid extraction; SRM: standard reference material; TFA: trifluoroacetic acid; TSQ: tandem sector quadrupole; UHPLC: ultra-high performance liquid chromatography; UHPSFC: ultra-high performance supercritical fluid chromatography; US-LLE: ultrasound assisted liquid–liquid extraction; UV: ultraviolet; VA-LLE: vortex-assisted liquid-liquid extraction.

However, for other samples with higher complexity, the application of multiple step methodologies has been found to be unavoidable so far. SLE and LLE have been applied to a wide variety of food matrices including milk and milk-based formulas, rice, cereals, bread, different kinds of fish, mushrooms, fruit juice, vegetables beverages, oil or eggs, among others [4,7,22–26,28,29,40,41]. In those cases, a great range of solvents such as hexane, ethyl acetate (EtOAc), MeOH, isooctane, chloroform, petroleum ether, diethyl ether, EtOH, pentane, cyclohexane, isooctane or IPA, as well as mixtures of them have been used, applying volumes in the range 1–100 mL and carrying out several repetitions in most cases, in order to get a quantitative extraction of the target analytes [4,7,28]. In almost all cases reported so far, the application of not only saponification [7,24–26,28,29,41], but also other hydrolysis treatments for the evaluation of cereals and nutritional formulas [23] or deproteinization for different milk samples [4,40] has been carried out, prior to extraction, to favor the cleaning of the samples and to remove interferences. With respect to the efficiency of these three strategies, despite saponification being the one most extensively applied, some studies have demonstrated the higher effectiveness of deproteinization in releasing vitamins from milk matrices [4,40]. In this sense, it is worth mentioning the work carried out by Gomes et al. [40], who developed a careful comparative study of both strategies for the effective extraction of eight vitamin D analogues (i.e., $D_3$, $D_2$, 25-OHD$_2$, 25-OHD$_2$, 24,25-(OH)$_2$D$_2$, 24,25-(OH)$_2$D$_3$, 1,25-(OH)$_2$D$_2$ and 1,25-(OH)$_2$D$_3$) from human, cow, mare, goat and sheep milk samples prior to their determination by HPLC-MS/MS. With this aim, 4 mL of sample was deproteinized using 8 mL of acetonitrile (ACN), vortexed for 2 min, incubated for 15 min at room temperature and subsequently centrifuged. For saponification, 4 mL of 8 M ethanolic KOH solution and 2.8 mL of 1 M ascorbic acid were added and, after vortexing for 2 min, the sample was hatched at room temperature for 12 h, before centrifugation. In both cases, a subsequent extraction of the obtained supernatant was carried out using 12 mL of hexane/DCM (1/4, *v/v*) twice, prior to determination. As can be seen in Figure 1, larger efficiency in terms of concentration was obtained for all target analytes using the deproteinization treatment, which demonstrated the higher suitability of this procedure, at least for the analysis of milk-based matrices.

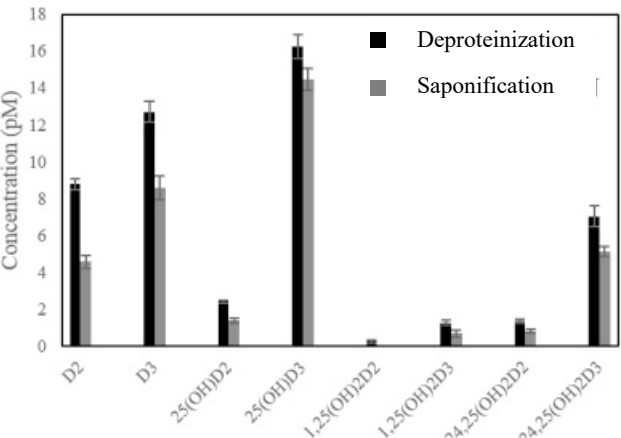

**Figure 1.** Comparison of deproteinization and saponification effectiveness for the release of vitamin D analogues from milk samples. Reproduced from [40], with permission from Elsevier, 2020.

Although in general terms, saponification, hydrolysis or deproteinization in combination with only one extraction technique have been found sufficient for the correct release of the analytes from the complex food samples, in some cases, the performance of additional treatments or extraction procedures has been necessary [24,25]. As an example, Mohanty et al. [25] developed a methodology for the extraction of $D_3$ from finfish and shellfish based on a modified Folch method using 450 mL of the mixture CHCl$_3$/MeOH (2/1, *v/v*) followed by alkaline saponification under reflux for 30 min and LLE of fish oil using 50 mL of petroleum ether prior to determination by HPLC-UV. The thorough extraction developed and clean extract obtained from the application of the described method allowed

the authors to determine $D_3$ and other fat-soluble vitamins in a great variety of fishes with a high fat content and, therefore, the evaluation of the contribution of these food commodities to the vitamin D daily intake in the population with an occurrence in the range 7.72–23.28 μg/kg in seventeen especies evaluated, including marine and freshwater fishes and moluscs.

However, despite the necessity of carrying out strong digestive treatments to get an efficient release of vitamin D and their analogues from food samples, there exists a remarkable trend to avoid them in order to simplify the procedures and keep the whole information about the matrix vitamin forms distribution in this kind of analysis [23]. In this regard, the fact is that in only a few cases, LLE and SLE have been directly applied for the evaluation of these analytes in food matrices [22,24]. Especially remarkable is the work carried out by David et al. [22], in which the authors applied a previously developed methodology [45], for the determination of vitamin D, as well as other fat-soluble vitamins in rice samples from different parts of Nigeria and Thailand by direct extraction of the matrix using 4 mL of hexane under $N_2$ steam for 5 min. After centrifugation, 1 mL of supernatant was evaporated, reconstituted in butanol and subsequently injected in the HPLC-UV system for the separation and determination of the target compounds. Although the Thailand species presented lower occurrence (13.8 mg/kg) than the Nigerian products (23.2–28.6 mg/kg), all species turned out to be poor sources of vitamin D, confirming the previous studies carried out in this area, in which the content of this vitamin has been associated with food commodities with a higher amount of fat such as fish, eggs and whole milk, among others [8–10].

Despite the fact that conventional extractions are the most commonly applied for the analysis of these analytes in food samples, the high amount of solvents used, as well as the complexity of the procedure, constitute clear disadvantages of their application. Apart from that, additional pre-treatments to carry out an effective extraction and clean-up of the sample, including hydrolysis or deproteinization, are still fundamental steps to achieve the successful analysis of compounds. For this reason, the search of new protocols that allow the miniaturization of such methods is an issue of special interest for the scientific community.

## 2.2. Solid-Phase Extraction

As shown in Table 3, SPE has been applied in this area for the extraction and/or sample clean-up of cereals and flour products [30], edible plants, seaweeds, fruit and fruit seeds [9,31], dietary supplements [31], porcine fat, meat and liver [42], cod liver supplements [8], beef, egg, chicken and fish [10]. The development of a previous digestion step, generally saponification, was the procedure most commonly carried out [9,10,30,42], although in some cases, the complexity of the evaluated samples also requires the performance of a solvent-based extraction between both stages [10,42], whereas in only one report, SPE has been applied directly after dilution of the sample using hexane [8]. In this last case, Bartoluccia et al. carried out a reliable methodology for the confirmation of the content of $D_3$ in cod liver oil-based supplements after the detection of $D_3$ intoxication in several consumers of this kind of pill. Authors pointed out the necessity of using a normal phase SPE with an apolar solvent eluent as the sample pre-treatment due to the lipidic nature of the sample since other alternatives, such as the application of $C_{18}$ sorbent in combination with polar eluent, have not demonstrated to yield an adequate release of fat-soluble vitamins from this type of sample. With this aim, 0.2% of the content of each capsule was diluted up to 1 mL with hexane and the sample was loaded onto a $NH_2$-propyl-SPE cartridge previously conditioned with the same solvent. After washing with hexane, the target compound was eluted with 1 mL of EtOAc. Finally, the extract was dried and the residue reconstituted with IPA prior to the separation and determination by HPLC-triple quadrupole (QqQ)-MS/MS. After validation of the methodology, the analysis of the samples showed a concentration of $D_3$ three orders of magnitude higher than the one indicated in the product (1.5 μg/capsule) for some of the batches evaluated. These results emphasize the necessity of the development of selective and reliable methodologies that allow the correct evaluation of all kinds of commercialized dietary supplements which guarantees food safety and protects consumer health.

**Table 3.** Recent applications of solid-phase extraction for the analysis of vitamin D-related compounds in food matrices.

| Analytes | Matrix (Amount) | Sample Pre-Treatment | Sorbent | Elution Solvents (Volume) | Determination Technique | Recovery % | LOD | Comments | Reference |
|---|---|---|---|---|---|---|---|---|---|
| $D_2$, $D_3$ | Cereal and flour-based products (1.5 g) | Saponification (KOH, EtOH, hexane, 60 °C, 30 min); vortex, centrifugation, upper layer drying, reconstitution and filtration | $C_{18}$ colum (-) | MeOH using a gradient at different flow rates | UHPLC-PDA | >70 | 0.05–0.12 mg/L | -Online SPE was carried out. -Vitamin K was also evaluated. -SRM was used for method validation. -Occurrence: 1.0–2.9 mg/kg. -LC conditions: $C_{18}$ column, mobile phase (MeOH/H$_2$O), 30 °C. | [30] |
| $D_2$, $D_3$, 25-OHD$_2$, 25-OHD$_3$ | Plants, seaweeds, fruit seed (-) | Saponification (KOH, ascorbic acid in H$_2$O, EtOH, overnight, room temperature) | Diatomaceous earth column (-) | Petroleum ether (2 × 30 mL) | HPLC-(QqQ)-MS/MS | 94–101 | 0.5 µg/kg | - ESI (+) was used as ionization source. -Deuterated compounds were used as ISs. -Derivatization was carried out using PTAD. -LC conditions: $C_{18}$ column, mobile phase (MeOH/H$_2$O), -. | [9] |
| $D_3$, 25-OHD$_3$ | Porcine fat and liver (0.2–1 g) | Saponification (KOH, sodium ascorbate, EtOH; overnight, room temperature); LLE (20% EtOAc in heptane); drying; reconstitution (1% IPA in heptane) | Silica column (500 mg) | 6% IPA in heptane; 10% IPA in heptane | HPLC-(QqQ)-MS/MS | 72–124 | <0.1 µg/kg [a] | -ESI (+) was used as ionization source. -Derivatization was carried out using PTAD. -LC conditions: $C_{18}$ column, mobile phase (MeOH/H$_2$O/FA/methyl amine), 50 °C. | [42] |
| $D_3$ | Fruit, fruit juices, dietary supplements (2 g) | SLE (MeOH); centrifugation; filtration; hydrolysis (carrez solutions) | $C_{18}$ column (500 mg) | 60% MeOH in water (3 mL); MeOH (3 mL); DCM (3 mL) | HPLC-DAD | 86–95 | 5.28 mg/L[b] | -9 more vitamins were also analyzed. -Occurrence: 2.5–74 µg/kg. -LC conditions: $C_{18}$ column, mobile phase (MeOH/H$_2$O/TFA), 30 °C. | [31] |
| $D_3$ | Cod liver oil-based dietary supplements (0.2% of each capsule) | Dilution in hexane (up to 1 mL) | NH$_2$-propyl column (200 mg) | EtOAc (1 mL) | HPLC-(QqQ)-MS/MS | - | - | -APCI (+) was used as ionization source. -Deuterated D$_3$ was used as IS. -Occurrence: 1.31 mg/pill. -LC conditions: $C_{18}$ column, mobile phase (MeOH/H$_2$O), -. | [8] |

**Table 3.** *Cont.*

| Analytes | Matrix (Amount) | Sample Pre-Treatment | Sorbent | Elution Solvents (Volume) | Determination Technique | Recovery % | LOD | Comments | Reference |
|----------|-----------------|----------------------|---------|---------------------------|-------------------------|------------|-----|----------|-----------|
| $D_3$, 25-OHD$_3$ | Pork, beef, egg, chicken, turkey, dolphinfish, salmon, tilapia (20 g) | Saponification (KOH, pyrogallol, EtOH; overnight, room temperature); LLE X 4 (petroleum ether/diethyl ether, 2/8, *v/v*); washing (5% KOH); drying ($N_2$); reconstitution (0.1% IPA in ciclohexane (0.5 mL)/DCM (0.6 mL)) | Silica colum (1000 mg) | 0.4–3.0% IPA in DCM | $D_3$: HPLC-DAD, 25-OHD$_3$: HPLC-(QqQ)-MS/MS | 73–88 | 0.4 µg/kg | -APCI (+) was used as ionization source. -D$_2$ and 25-OHD$_2$ were used as ISs. -CRM and ground pork control sample were used for method validation. -Poultry was also analyzed. -Further purification was carried out after SPE using normal phase semi-preparative HPLC. -Occurrence: D$_3$ (1.1–9.2 µg/kg); 25OHD$_3$ (0.9–3.6 µg/kg). -LC conditions: C$_{18}$ column, mobile phase D$_3$: (MeOH/H$_2$O),-; 25-OHD$_3$: (MeOH/H$_2$O/acetic acid), -. | [10] |

[a] LOQ, [b] instrumental LOD; 25-OHD$_2$: 25-hydroxyvitamin D$_2$; 25-OHD$_3$: 25-hydroxyvitamin D$_3$; ACN: acetonitrile; APCI: atmospheric pressure chemical ionization; C$_{18}$: octadecylsylane; CRM: certified reference material; D$_2$: vitamin D$_2$; D$_3$: vitamin D$_3$; DAD: diode array detection; DCM: dichloromethane; ESI: electrospray ionization; EtOAc: ethyl acetate; EtOH: ethanol; FA: formic acid; HPLC: high-performance liquid chromatography; IPA: isopropanol; IS: internal standard; LC: liquid chromatography; LLE: liquid–liquid extraction; LOD: limit of detection; MeOH: methanol; MS: mass spectrometry; MS/MS: tandem mass spectrometry; PDA: photodiode array detector; PTAD: 4-phenyl-1,2,4-triazoline-3,5-dione; QqQ: triple quadrupole; SLE: solid–liquid extraction; SPE: solid-phase extraction; SRM: standard reference material; TFA: trifluoroacetic acid; UHPLC: ultra-high performance liquid chromatography.

Regarding the type of sorbent applied in the different methods developed so far, it should by highlighted that, as indicated in the previous example, the use of silica as normal phase sorbent [10,42] has been commonly applied for the evaluation of samples with a high fat content such as porcine fat and liver [42], as well as meat, fish and eggs [10]. Indeed, the AOAC proposed in 1996 the AOAC 995.05 method [20] for the determination of cholecalciferol in infant formulas and enteral products using a silica-SPE after saponification and extraction using hexane, which pointed out the improvement with respect to previously developed methods in which LLE or SLE had been applied [19]. On the contrary, non-polar cartridges based on $C_{18}$ have been applied for the extraction of vitamin D analogues in cereal and flour products [30] or fruit-based commodities [31], and diatomeous earth for the evaluation of plants, seaweeds and fruit seeds [9]. The application of this sorbent in combination with 60 mL of petroleum ether as the elution solvent was used for the determination of $D_2$, $D_3$, 25-OHD$_2$ and 25-OHD$_3$ in several types of seaweeds and different parts of Australian plants like seeds, leaves or stems after saponification of the samples overnight at room temperature. The whole methodology was validated and applied for the determination of the target compounds with recovery values in the range 94–101% and a limit of detection (LOD) of 0.5 µg/kg. However, the authors remarked that the determination of 25-OHD$_3$ in roasted and milled seeds of wattleseeds could not be successfully performed due to the high amount of matrix interferences obtained in the final extract, thus demonstrating the inefficiency of the selected sorbent to carry out the release of 25-OHD$_3$ from this commodity.

As indicated at the beginning of this section, in some cases, the characteristics of the evaluated samples have made necessary the combination of SPE with other extraction and clean-up procedures to get an efficient extraction. Especially remarkable is the work carried out by Bilodeau et al. [10], in which the evaluation of $D_3$ and its metabolite 25-OHD$_3$ was carried out in pork, beef, egg, chicken, turkey, dolphinfish, salmon and tilapia, using a HPLC-diode array detector (DAD) for the quantification of $D_3$ and HPLC-(QqQ)-MS/MS for 25-OHD$_3$. Sample pre-treatment consisted of an initial alkaline saponification of the sample at room temperature using pyrogallol as the antioxidant agent, followed by LLE four times with 150 and 75 mL of petroleum ether/diethyl ether (2/8, *v/v*), washing with 5% KOH (one time) and water (twice), subsequent SPE, using 1 g of silica and 0.4–3% of IPA/DCM as the elution solvent, and semi-preparative HPLC using a normal phase column. Recovery values in the range 73–88% and an LOD of the method of 0.4 µg/kg demonstrated the suitability of the developed methodology. The reduction of matrix interferences and noise was considerable after de-application of the semi-preparative HPLC clean-up, especially for the determination of $D_3$ by UV, as can be seen in Figure 2, in which the HPLC-UV chromatograms of the separation between $D_3$ and the IS, used for its determination ($D_2$), in a solvent standard and in a real sample submitted to the developed methodology are shown.

As it has been demonstrated throughout the discussed literature, the application of SPE involves an improvement in the extraction and clean-up efficiency for the analysis of vitamin D analogues. In fact, the AOAC included in 1996 this procedure as alternative to SLE and LLE, commonly used with this aim. However, despite the mentioned advantages, it is true that the application of hydrolysis treatments and even previous solvent-based extractions are still necessary in many of the developed approaches, as well as the use of a high amount of organic solvents as the eluents. It is remarkable the low variety of sorbents tested in this area, which is an aspect that could considerably enhance the relevance of this technique, extending their application to a wider variety of matrices and improving the extraction and clean-up efficiency achieved.

## 2.3. Preparative Chromatographic Techniques

Preparative liquid chromatography has also been proposed as a suitable technique to get an adequate cleaning of the sample and to avoid the presence of interferences that could preclude the determination of vitamin D and its analogues in food matrices (see Table 4). In fact, the AOAC Official Method 2002.05 [21] describes the use of this technique for the evaluation of $D_3$ and $D_2$ in fatty food,

adding a UV detector at the outlet position that allows establishing the most accurate collection window and removing the highest amount of matrix interferences.

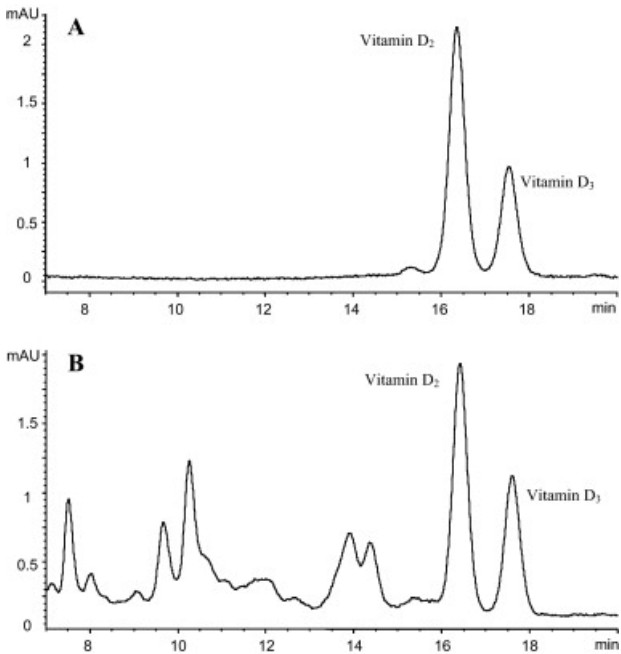

**Figure 2.** HPLC-UV chromatograms of a solvent standard (**A**) and in a real sample extracted using the developed methodology (**B**). Reproduced from [10], with permission from Elsevier, 2020.

Following the guidelines of the official method, all applications of the technique carried out so far have been performed using silica as the stationary phase for the evaluation of cacao-based food [32], yogurt, dietary supplements and margarine [33], fish and seafood [34], and orange juice [35] and applying an initial step of saponification followed by LLE, before LC separation. As an exception, Byrdwell et al. [35] carried out the determination of $D_3$ in fortified orange juice applying an LLE directly to the sample using ethyl ether and petroleum ether with ascorbic acid as the antioxidant agent and subsequent clean-up by preparative LC with a (25.0 cm × 4.6 mm, 5 mm) silica column using a mixture of IPA/methyl tert-butyl ether (MTBE)/cyclohexane/heptane (0.05/0.2/4.875/4.875, *v/v/v/v*) as the elution solvent. In this case, the particular characteristics of the sample, which presents a very low fat content in comparison with the rest of the analyzed matrices shown in Table 4, allowed the omission of a saponification step. The authors remarked that the application of LLE and preparative LC together with the use of $D_2$ as IS and the analysis by HPLC-UV and HPLC-MS constitutes the best combination of elements taken from the AOAC Official Methods 992.26 and 200.05 for the determination of $D_3$ in the selected matrix.

On the contrary, both Kühna et al. [32], who carried out the evaluation of cacao-based products, Nestola and Thellmann [33], who analyzed yogurt, dietary supplements and margarine, as well as Byrdwell et al. [34], who determined the content of vitamin D analogues in fish and seafood, described the necessity of applying a saponification step followed by LLE using hexane or a mixture of petroleum ether/diethyl ether to accomplish an adequate release of the analytes from the samples analyzed in each case. Especially remarkable is the evaluation of the influence of the saponification step carried out by Byrdwell et al., in which a comparison of of the amount of alkaline solution (30 and 60 mL of 50% KOH solution) effect on the reproducibility of the method was done, even though subsequently cleaning with the semi-preparative LC was applied. Results of the National Institute of Standards and Technology (NIST) standard reference material (SRM) of infant/adult nutritional formula and pulled salmon samples demonstrated that a stronger saponification increases the reproducibility between samples due to the increase in the matrix digestion [34].

**Table 4.** Recent applications of preparative chromatography for the analysis of vitamin D-related compounds in food matrices.

| Analytes | Matrix (Amount) | Sample Pre-Treatment | Column | Mobile Phase | Determination Technique | Recovery % | LOD | Comments | Reference |
|---|---|---|---|---|---|---|---|---|---|
| $D_2$, $D_3$ | Cacao, cacao-based food (-) | Saponification (KOH, -); LLE (hexane) | Silica column | - | HPLC-(QTrap)-MS/MS | - | - | -ESI (+) used as ion source. -Occurrence: 1.5–54.8 µg/kg. -Deuterated $D_3$ and $D_2$ were used as ISs. -Derivatization was carried out using PTAD. -LC conditions: $C_{18}$ column, mobile phase (ACN/$H_2O$/$NH_4$HCOO/FA), -. | [32] |
| $D_2$, $D_3$ | Yogurt, dietary supplement, margarine (-) | Homogeneization (hexane; water/EtOH (1/1, *v/v*)), saponification (KOH, 60 °C, 30 min); washing of hexane phase (water/EtOH (1/1, *v/v*)) | Silica column | 0.1% IPA in DCM | GC-(ToF)-MS | - | 50–150 pg | -EI used as ion source. -Comparison with HPLC-MS and HPLC-UV determination were carried out using different sample pre-treatments. -Online-HPLC-GC was performed reducing procedure complexity. -Deuterated $D_2$ and $D_3$ were used as ISs. -Determination was carried out using pyro-isomers due to the transformation reaction occuured at high temperature. | [33] |

**Table 4.** *Cont.*

| Analytes | Matrix (Amount) | Sample Pre-Treatment | Column | Mobile Phase | Determination Technique | Recovery % | LOD | Comments | Reference |
|---|---|---|---|---|---|---|---|---|---|
| D$_3$ | Finfish, shellfish (5–10 g) | Saponification (KOH, EtOH, ascorbic acid, 75 °C, 30 min, refluxing); LLE (petroleum ether/diethyl ether: AOAC 992.26) | Silica column | MeOH/ACN (2/8, *v/v*) | HPLC-DAD; HPLC-(TSQ)-MS | - | <2.0 µg/kg [a] | -APCI (+) used as ion source -D$_2$ used as IS. -Study of the improvement of the saponification process. -SRM was used for method validation. -Occurrence: (0.5–1.8 µg/kg) -LC conditions: C$_{18}$ column, mobile phase (MeOH/ACN), -. | [34] |
| D$_3$ | Fortified orange juice (30 mL) | LLE (Ethyl ether, petroleum ether, ascorbic acid) | Silica column | IPA/MTBE/ ciclohexane/heptane (0.05/0.2/4.875/4.875, *v/v/v/v*) | HPLC-UV-(TSQ)-MS, HPLC-(IT)-MS | - | UV: 0.298 µg/kg MS: 1.175 µg/kg | -APCI (+) was used as ionization source. -Occurrence: 10.7–16.63 µg/kg. -D$_2$ was used as IS. -LC conditions: C$_{18}$ column, mobile phase (MeOH/ACN), 40 °C. | [35] |

[a] LOQ; ACN: acetonitrile; AOAC: Association of Official Analytical Chemists; APCI: atmospheric pressure chemical ionization; C$_{18}$: octadecylsylane; D$_2$: vitamin D$_2$; D$_3$: vitamin D$_3$; DAD: diode array detection; DCM: dichloromethane; EI: electron impact; ESI: electrospray ionization; EtOH: ethanol; FA: formic acid; HPLC: high-performance liquid chromatography; IPA: isopropanol; IS: internal standard; IT: ion trap; LC: liquid chromatography; LLE: liquid–liquid extraction; LOD: limit of detection; MeOH: methanol; MTBE: methyl tert-butyl-ether; MS: mass spectrometry; MS/MS: tandem mass spectrometry; PTAD: 4-phenyl-1,2,4-triazoline-3,5-dione; QTrap: quadrupole linear-ion trap; SRM: standard reference material; ToF: time-of flight; TSQ: tandem sector quadrupole; UV: ultraviolet.

This procedure constitutes an adequate complement step to both solvent and sorbent-based extractions in order to improve the clean-up step, particularly in fat content samples for which normal stationary phases have shown good results. In fact, it could be considered as an alternative to the application of SPE since the use of off-line set-ups usually simplifies the procedures and reduces the consumption of solvents.

## 2.4. Miniaturized Techniques

The current trends in analytical chemistry are focused on the development of environmentally sustainable methodologies that enable a decrease in the consumption of harmful chemicals and diminish the impact of chemical activity on the environment. In this sense, miniaturized techniques that allow a reduction of cost, time and toxic and hazardous organic solvents consumption, as well as which require lower amounts of samples, are being increasingly used [46]. However, despite the undeniable advantages obtained by the application of miniaturized techniques and their wide application in other areas, their use for the determination of vitamin D and vitamin D analogues is strikingly low. As it is shown in Table 5, there are only a few reports in the area for the determination of these substances in food samples using magnetic-micro-dispersive solid-phase extraction (m-μ-dSPE) [37,39] and dispersive liquid–liquid microextraction (DLLME) [36,38] for the evaluation of milk, infant formulas, green vegetables, milk/fruit-based beverages and cereals. It is worth mentioning that only in one case, saponification was necessary prior to the application of the extraction process [36], whereas deproteinization was applied in the case of cereals and fruit/milk-based beverages [36,39].

A fast and simple methodology was developed by Viñas et al. [38] based on the application of DLLME for the evaluation of infant formulas (50–250 mg) and green vegetables (i.e., spinach and cos, iceberg and lamb's lettuce) (0.2–2.0 g). With this aim, an initial SLE using 3 mL of ACN as the extraction solvent was developed. Then, the extract obtained was used as the dispersant and mixed with 150 μL of carbon tetrachloride (extracting solvent). The mixture was rapidly injected in 6 mL of water and manually shaken for some seconds to favor the cloudy spread formation enabling the simultaneous preconcentration of the target analytes ($D_2$, $D_3$) and clean-up of the sample. After centrifugation, the small carbon tetrachloride drop (around 50 μL) containing the target compounds was taken from the bottom of the tube, dried and reconstituted in the same volume of ACN prior to injection in a HPLC-ion trap (IT)-MS. The good recovery values obtained in the range 88–103% and the LODs between 3.1 and 4.0 μg/L demonstrated the efficiency of the technique for the evaluation of the complex samples selected. In addition, the method demonstrated the simplification of the procedure in comparison with methods described until now and a reduction in the extraction time and organic solvents consumption.

**Table 5.** Recent applications of miniaturized extraction techniques for the analysis of vitamin D-related compounds in food matrices.

| Analytes | Matrix (Amount) | Sample Pre-Treatment | Technique | Determination Technique | Recovery % | LOD | Comments | Reference |
|---|---|---|---|---|---|---|---|---|
| $D_3$ | Wheat flour, powder bread (2.0 g) | Saponification ($H_2O$, KOH, EtOH, sodium ascorbate; -) (ultrasounds, 5 min, 25 °C); deproteinization (carrez solutions) | DLLME (dispersant: EtOH (650 µL); extractant: octanol (80 µL)) | HPLC-UV-Vis | 87–94 | 0.7 µg/kg | -LC conditions: $C_{18}$ column, mobile phase (MeOH), 25 °C. | [36] |
| $D_2$, $D_3$ | Milk (1 mL) | Dilution (phosphate buffer, 9 mL) | m-µ-dSPE (sorbent: $Fe_3O_4$@PPy (30 mg); elution solvent: ACN (1 mL)) | HPLC-UV | 72–90 | 0.02–0.05 µg/L | -Occurrence: $D_2$: 6.5–10.2 µg/L; $D_3$: 2.7–8.8 µg/L. -LC conditions: $C_{18}$ column, mobile phase (MeOH/ACN), -. | [37] |
| $D_2$, $D_3$ | Infant formula (50–250 g), green vegetables (0.2–2 g) | SLE (ACN, 3 mL) | DLLME (dispersant: ACN (3 mL); extractant: $CCl_4$ (150 µL)) | HPLC-(IT)-MS | 88–103 | 3.1–4.0 µg/L | -APCI (+) was used as ionization source. -Vitamin $K_1$, $K_2$ and $K_3$ were also evaluated. -CRM were used for method validation. -Different LC columns were evaluated. -DAD was also applied as detector. -LC conditions: $C_{18}$ column, mobile phase ($ACN/H_2O$/IPA), 25 °C. | [38] |
| $D_2$ | Fruit juice, milk beverage (0.1 mL) | Deproteinization (ACN, 0.9 mL) | m-µ-dSPE (sorbent: $C_{18}$@$Fe_3O_4$ (30 mg); elution solvent: ACN (-)) | CLC-UV | | | -Other fat-soluble vitamins were also evaluated. -LC conditions: $C_{18}$ column, mobile phase (ACN), -. | [39] |

ACN: acetonitrile; APCI: atmospheric pressure chemical ionization; $C_{18}$: octadecylsylane; CLC: capillary liquid chromatography; CRM: certified reference material; $D_2$: vitamin $D_2$; $D_3$: vitamin $D_3$; DAD: diode array detection; DLLME: dispersive liquid–liquid microextraction; EtOH: ethanol; HPLC: high-performance liquid chromatography; IPA: isopropanol; IT: ion trap; LC: liquid chromatography; LOD: limit of detection; MeOH: methanol; m-µ-dSPE: magnetic-micro-dispersive solid-phase extraction; MS: mass spectrometry; PPy: polypyrrole; SLE: solid–liquid extraction; UV: ultraviolet.

The use of nanomaterials as sorbents in sample pre-treatments of food matrices is a strategy widely applied due to the particular characteristics of such materials that allows the miniaturization of the extraction and clean-up procedures, minimizing the sample and solvent amount requirements, simplifying the methodologies and providing very good efficiencies with a minimal environmental impact. Particular interest has reached the magnetic nanomaterials. Those not only provide a high surface area, increasing the active point, in which the sorbent can interact with the substances of interest and present high tuneable capacity, favoring the versatility and specificity of the extractions as occurs with the rest of nanomaterials, but their use also simplifies even more the procedures as a result of their magnetic properties, which allows the easy separation of the sorbent and the sample using an external magnet [47]. These advantages were precisely the ones provided by the sorbents developed by Jiao et al. [37] and Hu et al. [39] for the determination of vitamin D using m-μ-dSPE. Particularly simple was the methodology described by Jiao et al. [37] using polypyrrole (PPy)-coated $Fe_3O_4$ nanoparticles (PPy@$Fe_3O_4$) to carry out the extraction and preconcentration of $D_3$ and $D_2$ from milk samples prior to their determination by HPLC-UV. With this aim, 1 mL of sample was initially diluted with 9 mL of buffer phosphate solution. Then, the m-μ-dSPE was directly performed without any additional step by using 30 mg of PPy@$Fe_3O_4$. After stirring for 10 min, the sorbent was magnetically separated from the sample solution with the assistance of a magnet and then washed with 2 mL of water. Finally, the analytes were desorbed with 1 mL of ACN by ultrasonication for 5 min. The sorbent was again separated using an external magnet and the eluate dried, reconstituted and submitted to HPLC-UV analysis. The good recovery values obtained in the range 72–90% and the low LODs (0.02–0.05 μg/L) demonstrated the effectiveness of the π–π interactions between the sorbent commodities and the target analytes, as well as the good performance of the developed methodology with a pre-treatment time of only 15 min and avoiding the digestion steps.

The advantages of these techniques in terms of simplicity and chemicals consumption are undeniable and the great variety of sorbent and solvent materials that could be applied increases exponentially the range of application of these methods. However, their application to solid samples still involves the use of an initial pre-treatment using a high volume of solvents which could be considered as a disadvantage of their application. In this sense, only a few approaches have been applied so far. The application of other miniaturized procedures such as matrix solid phase dispersion could be an interesting alternative to solve this problem.

## 3. Recent Applications of Chromatographic Techniques

In the last thirty years, the use of chromatographic techniques for the separation and determination of vitamin D and its analogues has become a common trend in this area due to the advantages that these techniques offer with respect to bio-assay methods, particularly the possibility of carrying out the analysis of individual analytes, providing an accurate evaluation of the levels of each analogue in the organism [10].

Regarding the most recent applications, as shown in Tables 1–4, the combination of LC with UV or MS detection have been the most common strategies [3], whereas gas chromatography (GC) applications have been hardly reported due to the easy isomerization of vitamin D analogues at high temperature [3], which would preclude the individual determination of this group of analytes. The application of other chromatographic techniques in the last ten years has been reduced to the use of supercritical fluid chromatography (SFC) hyphenated with MS. However, it has been applied in few cases owing to the current stage of development of this technique in terms of commercial instrumentation, etc. [48].

### 3.1. Liquid Chromatography Applications

Regarding the use of LC, different modalities have been applied, including HPLC [7–10, 22,24,25,27–29,31,32,34–38,40,42], two dimensional-HPLC (2D-HPLC) [41], ultra-high performance liquid chromatography (UHPLC) [26,30] and capillary liquid chromatography (CLC) [38], although,

among them, HPLC has been the most extended as a consequence of its higher availability in most laboratories. Aqueous–organic mixtures with ACN and MeOH, fundamentally, as well as IPA [7–10,22,24,26,29–32,34,38,40–42] or pure organic solvents and mixtures of them have also been used [25,27,28,35–37,39] as mobile phases without [8–10,22,25,27,28,30,34–39] or with additives including formic acid (FA) [7,26,40–42], ammonium formate [24,29] or both of them [32], acetic acid [10], methyl amine [42] and trifluoro acetic acid (TFA) [31], especially for LC-MS hyphenations. With respect to the columns usually selected for this type of application, despite the fact that the use of normal phases has been previously reported for the analysis of these compounds, the latest publications describe, fundamentally, the separation using reversed-phase columns with very good results. However, it is worth mentioning that in many cases, a clean-up step using polar sorbents in SPE or normal phases in preparative chromatography has been necessary in order to remove lipid interferences, especially for high-fat content sample analysis [1].

The combination of LC with UV detection or DAD was also very common in the previous decades. However, the high matrix effect that precludes the correct determination of these vitamins in food samples, as well as the limitations in terms of sensitivity and selectivity have brought about the necessity to move on to the use of more suitable systems such as LC-MS or LC-MS/MS which have become the gold tools for the simultaneous determination of vitamin D analogues [3]. In this sense, UV has been used for the determination of only one or two analytes simultaneously ($D_2$ and $D_3$) in most of cases [22,25,28,30,31,36,37,39,41] using, fundamentally, 265 nm as the most suitable wavelength, and their combination with MS has been necessary in some cases to reach the levels at which some analytes are present in the samples analyzed [10]. As a remarkable exception, Wittig et al. [7] carried out the determination of ten vitamin D analogues (i.e., $D_2$, $D_4$, previtamin-$D_2$ (Pre-$D_2$), previtamin-$D_4$ (Pre-$D_4$), tachysterol$_2$, tachysterol$_4$, lumisterol$_2$, lumisterol$_4$, ergosterol, 22,23-dihydroergostero), as well as $D_3$ used as the surrogate, in oyster mushrooms after UV-B treatment by HPLC-UV. As can be seen in Figure 3, clean chromatograms of the evaluated samples treated with hot alkaline saponification followed by LLE and subsequent separation by HPLC were obtained with an acceptable separation of all analytes. The good recovery (98%) and the good sensitivity of the developed methodology (LODs: 0.02–0.06 mg/kg) enabled its successful application to the monitoring of these compounds in mushroom samples submitted to UV-B exposition in order to enhance their content in vitamin D-active metabolites with therapeutic functions. However, although analytes quantification was performed using UV detection, it should be highlighted that MS determination was also carried out to confirm their identification.

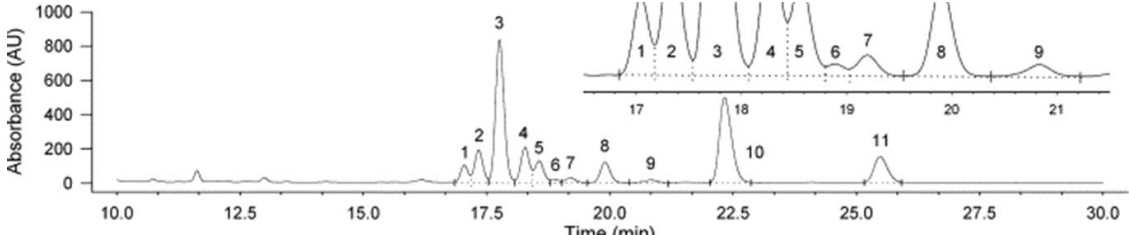

**Figure 3.** HPLC-UV chromatogram of an oyster mushroom sample submitted to saponification and subsequent LLE after UV-B treatment. Analytes identification: (1) pre-$D_2$; (2) tachysterol$_2$; (3) $D_2$; (4) $D_3$ (IS); (5) lumisterol$_2$; (6) pre-$D_4$; (7) tachysterol$_4$; (8) vitamin $D_4$; (9) lumisterol$_4$; (10) ergosterol (provitamin $D_2$); (11) 22,23-dihydroergosterol (provitamin $D_4$). Adapted from [7], with permission from Elsevier, 2020.

The development of methodologies based on the coupling of LC with MS systems has been carried out in combination with all kinds of sample pre-treatments previously described as a consequence of the advantages that this strategy provides in terms of sensitivity and selectivity. Both MS detection [27,34,35,38] and MS/MS [7–10,24,26,27,29,41,42] have been applied using QqQ in most

cases [7–10,24,26,29,32,40–42]. Nevertheless, other analyzers such as tandem sector quadrupole (TSQ) [34,35], quadrupole linear-ion trap (QTrap) [32] and IT [35,38] have also been used recently.

Regarding the ionization sources, both electrospray ionization (ESI) [9,24,26,29,32,40–42] and APCI [7,8,10,27,34,35,38] in positive mode have been applied, using a great variety of additives as mentioned at the beginning of this section. However, FA has been preferred in the majority of cases for both modes.

Other usual practice in the evaluation of this type of compound by MS is the application of a derivatization step in order to increase the sensitivity, especially for those analogues which are present at a lower concentration and when the amount of sample available also is small. In recent years, 4-phenyl-1,2,4-triazoline-3,5-dione (PTAD) has been the chosen reagent in all cases in the area of food analysis since it presents a high effectiveness, increasing sensitivity between 7 and 100 times depending on the compound [40].

Finally, it should also be highlighted the use of deuterated [8,9,24,26,32,40,41] and non-deuterated [7,10,27,34] surrogates and ISs for the validation of a great number of the evaluated matrices using not only MS, but also UV or DAD detectors in order to guarantee the reliability of the obtained results.

### 3.2. Gas Chromatography

GC has been hardly applied as a separation technique for the analysis of vitamins in food matrices due to the low stability of these compounds at high termperature [33]. This was pointed out by Nestola and Thellmann in their work reported in 2015, in which the analysis of $D_2$ and $D_3$ was carried out using an online hyphenation between a preparative HPLC and GC-time of flight (ToF)-MS, after saponification, in yogurt, dietary supplements and margarine [33]. Authors highlighted the simplicity of the online procedure and compared it to HPLC-MS/MS, indicating that similar results were obtained with less manual clean-up performance. However, the isomerization of the analytes as a consequence of the high temperature applied in the GC separations could not be avoided, generating double peaks which hindered the correct identification and quantification of vitamin D. This constraint made necessary to carry out the determination of the target analytes using their isomerized pyro-structures in combination with deuterated ISs by a "cross-contribution" procedure. Despite the advantages that the online procedure provided, the limitations of using GC for the analysis of thes kinds of substances are evident, which explains the higher use of other techniques such as LC.

### 3.3. Supercritical Fluid Chromatography

SFC is a separation technique which integrates the characteristics of LC and GC. Developed in the 1960s, it did not gain popularity until the 1980s due to infrastructure limitations. Since 2009, new nstruments that allow the hyphenation of the technique with different detectors, as well as other improvements such as a reduction of dead volume, higher control of pressure or the use of stationary phases with smaller particle size have been developed by different companies, which has increased the number of applications and publications using this technique. SFC is based on the separation of the analytes using an eluent based on liquid and/or supercritical carbon dioxide mixed with a co-solvent or solvent mixture. Carbon dioxide is used as the eluent almost always owing to its particular advantages in terms of low toxicity, availability and accessible critical point, among others. The separation of analytes not only depends on the identity of the stationary phase ligand, the nature of the eluent constituents and how these are mixed in the gradient elution, but also on the density of the mobile phase which has an important influence on analytes solubility. The customized modification of such parameters allows the comprehensive separation of a wide variety of analytes with different polarities and also the selective separation of compounds with similar structures, as occur with enantiomers. The large range of polarities that cover this technique involving all compounds with Log P between −2.11 and −10.12 makes fat-soluble vitamins (Log P 6–9) ideal candidates for their separation using SFC [3,48–51].

Despite the recent development of equipment that guarantee the robustness and efficiency of this technique [3], in the area of food analysis and, particularly, for the evaluation of vitamin D analogues, the application of SFC is limited to a few studies developed in recent years for the evaluation of milk [4] and infant and adults nutritionals, as well as cereal samples [23]. In both cases, the hyphenation of ultra-high-performance supercritical fluid chromatography (UHPSFC) with MS has been applied using a particle size lower than 2 μm in order to improve the efficiency and rapidness of the technique, as well as decrease the consumption of solvents and reduce void volumes [51,52]. QqQ as analyzer and APCI ionization source in positive mode have been chosen on both publications. Besides, as a consequence of the particular characteristics of the mobile phase used in this technique, the coupling with MS involves the application of a make-up solvent added to favor the transfer of the compounds to the MS and guarantee the determination of the analytes. This solvent presents an important influence on the sensitivity of the methodology. In these cases, ammonium formate in MeOH was selected to achieve an adequate ionization of the analytes including $D_2$ and $D_3$ [4,23], as well as hydroxylated metabolites [4]. As described for LC separations, derivatization using PTAD was also selected for the determination of these analogues with low levels of concentration in the samples of interest [4,23].

Column selection is considered a big challenge in SFC since the specific interactions established between the analytes and stationary phases are important to achieve the successful separation of vitamin D isomers. In this sense, fluoro-phenyl columns with charged surface hybrid (CSH) particles [4] and 1-aminoanthracene (1-AA) columns [23] have been the ones selected so far. An interesting study about column selection was the one carried out by Oberson et al. [4]. In order to achieve the separation of PTAD-derivatized $D_2$, $D_3$, 25-OHD$_2$, 3-epi-25-OHD$_2$, 25-OHD$_3$ and 3- epi-25-OHD$_3$, the authors checked nine different columns with a great variety of stationary phases including fluoro-phenyl-CSH, ethylene bridged hybrid (BEH)-C$_{18}$, BEH-2-ethylpyridine (2-EP), 1-AA-C$_{18}$, high-strength silica (HSS)-C$_{18}$, Diol-C$_{18}$, 2,6-cholesteryl-C$_{18}$, C$_{30}$ and amylose tris(3,5-dimethylphenylcarbamate) (AD-3) columns. Most of them provided a poor separation of the analytes and low sensitivity, except in the cases of the fluoro-phenyl-CSH and AD-3 columns. However, the fluoro-phenyl-CSH column, using MeOH and water as the cosolvents and ammonium formate as the additive at 45 °C, was finally selected for the validation of the whole methodology. As can be seen in Figure 4, the baseline separation of isomers was obtained with better peak shape than the results obtained using the AD-3 phase, although double peaks were observed for the 3-epi-25-OHD forms.

Finally, it should also be highlighted that mixtures of water and MeOH using ammonium formate as the additive were selected in the articles recently published in the area of vitamin D analysis in food matrices. The higher sensitivity provided by MeOH as compared with ACN has been demonstrated previously by some studies developed for the determination of these compounds by SFC-MS [4].

Although the application of this technique is still scarce, the great comprenhensivity of SFC and the excellent sensitivities obtained when it is hyphenated with MS systems makes this approach one of the most promising techniques for the analysis of vitamin D and its metabolites in food matrices.

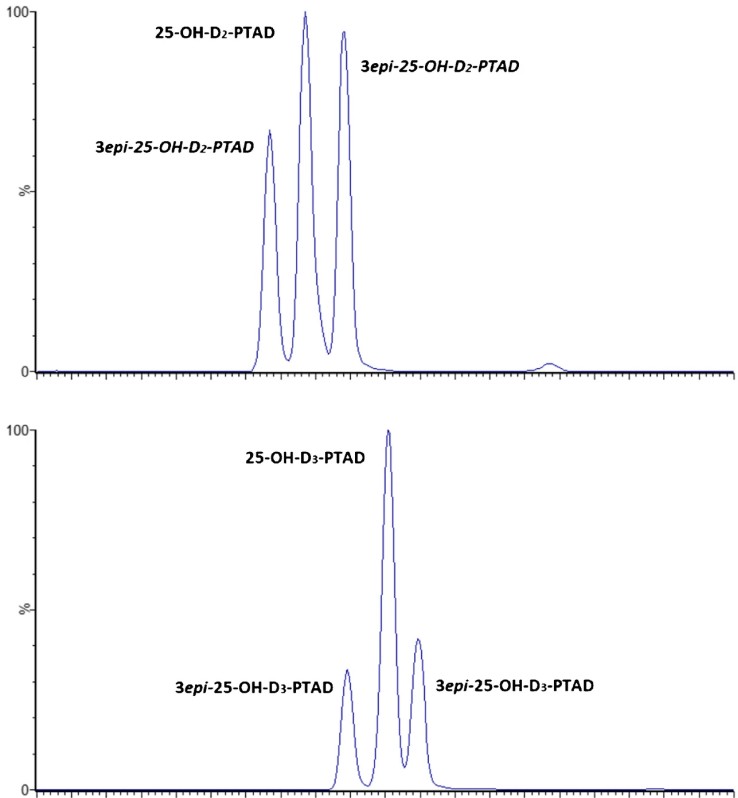

**Figure 4.** UHPSFC-MS/MS chromatograms of the separation of PTAD-derivatized 25-OHD$_2$, 3-epi-25OHD$_2$, 25-OHD$_3$ and 3-epi-25OHD$_3$ using fluoro-phenyl column-CSH (1.7 µm, 3 × 100 mm). Gradient of MeOH/water (98/2, *v/v*) containing 10 mM ammonium formate as organic modifier on CO$_2$. Column temperature: 45 °C. Atmospheric back pressure regulator: 128 bar. Make-up solvent was 10 Mm ammonium formate in MeOH, flow rate 0.4 mL/min. Reproduced from [4], with permission from Springer, 2020.

## 4. Conclusions and Future Remarks

The important role that vitamin D and its metabolites play in several vital functions for human beings and the influence that their status in the organism has on the development of a great number of diseases have made vitamin D deficiency a global health concern. The low subcutaneous bio-production of vitamin D, especially for populations at high latitudes during winter months, makes the diet the main source of these compounds. For this reason, the determination of vitamin D and vitamin D analogues contents in commonly consumed food items, as well as the identification of food products that are especially rich in those substances, is of uttermost importance. Additionally, each time, the most common use of supplements or fortified foods that guarantee the adequate levels of these compounds in the population also urges a reliable assessment of these commercialized products for ensuring food safety.

All the abovementioned makes evident the necessity of developing reliable methodologies that enable an accurate determination of the content of vitamin D in foodstuffs. In this sense, chromatographic systems have become the most suitable techniques to carry out the individual evaluation of vitamin D and its analogues in food matrices displacing other methods initially applied such as bio-assays. Particularly, the hyphenation of different modalities of LC with conventional UV detectors and, most recently, with MS systems have been the most recurrent strategies due to the advantages that this technique presents with respect to GC in which analytes' isomerization occurs at high temperature. In addition, the commercialization of robust SFC systems and the advantages that this technique provides in terms of comprehensiveness, selectivity and peak capacity has allowed the

performance of suitable methodologies based on the use of supercritical fluids, which have become a promising alternative for the evaluation of vitamins.

Nevertheless, the great complexity of food matrices makes sample analysis difficult without pre-treatment. In this sense, the application of digestive treatments, such as saponification or deproteinization, is frequent in most applications. Those are usually combined with conventional SLE or LLE. However, other widely known techniques, such as SPE or preparative LC, have also been applied in recent years with considerable improvements in the efficiency of clean-up performance. However, the current trends in analytical chemistry based on the developed methodologies supported by the principles of green chemistry have come up with the application of miniaturized techniques also in this field, reducing the costs and time, as well as the sample and solvent amount requirements of conventional methodologies. So far, the techniques applied, DLLME and μ-dSPE, have provided promising results, although their use for the analysis of solid samples still requires previous pre-treatments that involve the use of high volumes of organic solvents. In this sense, it could be interesting the use of other approaches more suitable for this kind of sample such as matrix solid-phase extraction that would allow the direct application in the matrix and the accomplishment of green chemistry principles.

Ultimately, it is evident that there is much to be done in the field. There are limitations that hamper the individual and simultaneous determination of the different kinds of analogues including $D_3$, $D_2$, precursor forms, hydroxylated metabolites and ester analogues, for which there is scarce information up to date. Besides, the effective release of the analytes from the complex food samples without matrix interferences is still a big challenge for the scientific community. However, the current developments focus on the combination of simple, fast and effective sample pre-treatment with the state-of-the-art chromatographic equipment and the most sensitive and versatile detection systems constituting the most promising alternatives to achieve the goals posed in this area.

**Author Contributions:** All Authors contributed equally to the present work. All authors have read and agreed to the published version of the manuscript.

**Funding:** This research was funded by the Carl Trygger Foundation (project: CTS 18:395).

**Acknowledgments:** B.S.-R. would like to thank Carl Trygger Foundation for funding her postdoctoral stipend. This work was supported by the Carl Trygger Foundation (project: CTS 18:395).

**Conflicts of Interest:** The authors declare no conflict of interest.

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
