# Peer review of "Recent Advances in the Analysis of Vitamin D and Its Metabolites in Food Matrices"

_separations, doi:10.3390/separations7020036_

Round 1

Reviewer 1 Report

This review is focused on the analytical techniques for the determination of vitamin D and its metabolites in food matrices. The subject of the article is of great interesting from both food science and analytical chemistry point of views. An exhaustive review of the extraction techniques as well as chromatographic methodologies is reported.

The paper is well-structured and the objective is clearly defined.

However, the manuscript could improve with some considerations:

- Including the chemical structure of vitamin D, its analogues and metabolites, would help illustrate the text.

- Including a table with different foods and their vitamin D content would be advisable.

- In conclusions section the advantages and drawbacks of analytical methods would be recommended to comment

I do express my positive opinion on the acceptance of the manuscript to be published after minor revisión.

Reviewer 2 Report

Recent advances in the analysis of vitamin D and its 2 metabolites in food matrices

by Bárbara Socas Rodríguez, Margareta Sandahl, Cecilia Holm  and Charlotta Turner

This review describes the analytical approaches in the analysis of Vitamin D in foods.
Vitamin D is perhaps one of the most supplemented vitamins whether by means of dietary supplements or in enriched and functional foods. Analytical methods to control and define its content in foods are focused to evaluate the compliance with claimed effect on foods, for its efficacy and on the other hand, to avoid adverse effect due to a high concentration in foods.
The paper is well written and composed coherently, following a scientific logic pathway facilitating reading. Furthermore, it is not boring because it provides practical aspects that and useful information for researcher that have to work on this topic.
In my opinion the manuscript can be published as such.

Reviewer 3 Report

This is an interesting, well organized and well written review on determination of vitamin D and its analogues in foods, therefore I recommend the publication after minor modifications.

I have only a few remarks:

  • Please reduce a little bit the conclusions and future remarks avoiding repetitions throughout the text.
  • Include a little paragraph on GC determination, also including the use of coupled chromatographic techniques such as on-line HPLC-GC-MS (which has been successfully applied by Nestola and Thellmann in 2015)
  • Line 19: “novel sample pre-treatments has become in a suitable strategy”, change with “novel sample pre-treatments has become a suitable strategy”
  • Line 23: “used in the field”: please specify better or delete
  • Line 73: “However, overdoses must be avoided.” The concept has been already reported. I would delete this sentence.

Reviewer 4 Report

This review is well written and thorough.  I recommend publication as is with minor copy editing.  Since it is a review of other work, I gave only 4 stars on contribution. 
